# Heat Shock Protein 27 Is Involved in the Bioactive Glass Induced Osteogenic Response of Human Mesenchymal Stem Cells

**DOI:** 10.3390/cells12020224

**Published:** 2023-01-05

**Authors:** Laura Hyväri, Sari Vanhatupa, Miina Ojansivu, Minna Kelloniemi, Toni-Karri Pakarinen, Leena Hupa, Susanna Miettinen

**Affiliations:** 1Adult Stem Cell Group, BioMediTech, Faculty of Medicine and Health Technology, Tampere University, Arvo Ylpön katu 34, 33520 Tampere, Finland; 2Research, Development and Innovation Centre, Tampere University Hospital, Elämänaukio, Kuntokatu 2, 33520 Tampere, Finland; 3Department of Plastic and Reconstructive Surgery, Tampere University Hospital, Elämänaukio, Kuntokatu 2, 33520 Tampere, Finland; 4Regea Cell and Tissue Center, Tampere University, Arvo Ylpön katu 34, 33520 Tampere, Finland; 5Johan Gadolin Process Chemistry Centre, Åbo Akademi University, Henrikinkatu 2, 20500 Turku, Finland

**Keywords:** mesenchymal stem cells, osteogenesis, bioactive glass, p38/HSP27 signaling, phosphorylation

## Abstract

Bioactive glass (BaG) materials are increasingly used in clinics, but their regulatory mechanisms on osteogenic differentiation remain understudied. In this study, we elucidated the currently unknown role of the p38 MAPK downstream target heat shock protein 27 (HSP27), in the osteogenic commitment of human mesenchymal stem cells (hMSCs), derived from adipose tissue (hASCs) and bone marrow (hBMSCs). Osteogenesis was induced with ionic extract of an experimental BaG in osteogenic medium (OM). Our results showed that BaG OM induced fast osteogenesis of hASCs and hBMSCs, demonstrated by enhanced alkaline phosphatase (ALP) activity, production of extracellular matrix protein collagen type I, and matrix mineralization. BaG OM stimulated early and transient activation of p38/HSP27 signaling by phosphorylation in hMSCs. Inhibition of HSP27 phosphorylation with SB202190 reduced the ALP activity, mineralization, and collagen type I production induced by BaG OM. Furthermore, the reduced pHSP27 protein by SB202190 corresponded to a reduced F-actin intensity of hMSCs. The phosphorylation of HSP27 allowed its co-localization with the cytoskeleton. In terminally differentiated cells, however, pHSP27 was found diffusely in the cytoplasm. This study provides the first evidence that HSP27 is involved in hMSC osteogenesis induced with the ionic dissolution products of BaG. Our results indicate that HSP27 phosphorylation plays a role in the osteogenic commitment of hMSCs, possibly through the interaction with the cytoskeleton.

## 1. Introduction

Mesenchymal stem cells (MSCs) are the multipotent precursor cells playing a central role in bone development and regeneration [1,2,3], making them intriguing candidates for bone tissue engineering (TE) applications. MSCs can be obtained from adult tissues, including bone marrow and the stromal vascular fraction (SVF) of adipose tissue [4,5]. Additionally, bioactive glass (BaG) materials have been extensively studied during the past decades in the context of bone tissue engineering (TE), because of their ability to stimulate osteogenesis and bone growth [6,7]. In a clinical setting, bone regeneration is already aided by well-established silicate BaGs such as 45S5 (Bioglass^®^), or in non-load-bearing sites BaG particulates, such as S53P4 (BonAlive^®^) granules [8]. Novel glass compositions have been developed in search of improved biological responses, and the incorporation of metal ions such as Cu, Co, Mn, Sr, Mg, Zn, Li, and B to the BaGs has been proven advantageous for osteogenic induction *in vitro* [9,10,11]. The ionic products released from the BaGs have been found to promote proliferation [12] and the osteogenic differentiation of stem cells [13,14,15,16]. However, the molecular details of how BaGs modulate cell differentiation remain understudied.

Mitogen activated protein kinase (MAPK) family member p38 is involved in the skeletal development and bone homeostasis [17,18,19]. Due to the versatile involvement of p38 MAPK in many cellular processes besides cell differentiation, the details of these signaling events have been extensively studied, as reviewed by Canovas and Nebreda [20]. The pathway is activated by environmental stresses, cytokines, and mechanical stimuli [20], and there are reports showing that p38 MAPK signaling is activated by BaGs [21,22]. 

*In vitro*, osteogenesis is accompanied by the physical change of the cell cytoskeleton [19,23,24,25], and the cellular mechanisms regulating the cytoskeletal dynamics are significant in regulating the osteogenic lineage commitment of hMSCs [19,24,26,27,28]. The p38 MAPK downstream target heat shock protein 27 (HSP27), also known as heat shock protein B1 (HSPB1), is a chaperone suggested to be involved in the control of actin dynamics [17,29,30,31]. Therefore, we hypothesized that HSP27 would be involved in the BaG induced osteogenic fate of human mesenchymal stem cells (hMSCs). 

Transient upregulation of HSP27 (or its murine analogue HSP25) has been reported in the *in vitro* differentiation programs of many cell types, as summarized previously [31]. Importantly, there is some prior evidence that HSP27 is involved in osteogenesis. Shakoori and co-workers showed elevated HSP27 gene expression during osteogenesis of rat osteoblasts [32], and the expression was also linked to osteogenesis of electrically stimulated human MSCs [33]. Additionally, immunohistochemical analyses have shown spatial activation of HSP27 during bone development in human fetal craniofacial tissues [34], and in rat tibiae [35]. 

Phosphorylation and oligomerization are important characteristics of heat shock proteins and have been proposed to guide their functions [30,36]. Unphosphorylated HSP27 exists mainly in the cytosol as large oligomers. The p38 MAPK substrate MAPK-activated protein (MAPKAP) kinase 2 (MK2) reversibly phosphorylates HSP27 on mainly Ser-78 and/or Ser-82 residues, changing its conformation into dimers. According to previous studies, this conformation promotes the nuclear and actin-related localization of HSP27. [17,36,37] We aimed to uncover the phosphorylation status of HSP27 during osteogenesis of hMSCs and reveal whether HSP27 could be involved in the osteogenic differentiation fate through its ability to interact with the cytoskeleton. 

In this work, we studied the involvement of p38/HSP27 signaling in hMSC BaG induced osteogenesis. To achieve this, we cultured human adipose stem cells (hASCs) and human bone marrow stem cells (hBMSCs) harvested from adult tissues in basic medium (BM), osteogenic medium (OM), and basic medium with ionic dissolution products (Ca, K, Mg, Si, Na, and B) released from an experimental silica-based glass 3-06 [14] (BaG BM), or with the ionic dissolution of BaG in OM (BaG OM). The ion concentrations of the ionic dissolution were determined previously as [mg/kg]: Ca, 131; K, 172; Mg, 16; Si, 56; Na, 3750; and B, 2.6; P was below the limits of quantification [14]. The utilization of BaG extract allowed us to examine the effect of the bioactive ions without the cell-biomaterial contact and subsequent activation of cell adhesion -related intracellular cascades. 

We analyzed the osteogenic potential of the cells with alkaline phosphatase activity, using the Alizarin Red mineralization assay and immunocytochemical (ICC) staining of collagen type I. We studied the intracellular protein activation of p38 MAPK, MK2, and HSP27 using Western Blotting of phosphorylated and basal proteins. The relevance of HSP27 phosphorylation on the osteogenic outcome was studied with the p38 MAPK inhibitor SB202190. Lastly, the intracellular localization of HSP27, pHSP27(S78), and Phalloidin-stained F-actin were studied under p38 MAPK inhibition. This study provides new knowledge on the role of p38 MAPK signaling in hMSC osteogenesis through a downstream target HSP27, and its ability to interact with the actin cytoskeleton. 

## 2. Materials and Methods

### 2.1. Isolation and Expansion of hASCs and hBMSCs

The hASCs were isolated from donated subcutaneous abdominal adipose tissue samples collected at the Tampere University Hospital Department of Plastic and Reconstructive Surgery, from three female donors aged 53, 60, and 63. Isolation of the hASCs was performed as described previously by Lindroos and co-workers [38]. Briefly, the adipose tissue was digested mechanically and enzymatically (Collagenase type I; Thermo Fisher Scientific, Waltham, MA, USA) and centrifuged and filtered to separate the stem cells. The isolated hASCs were cultured on Nunclon Delta surface polystyrene culture flasks (Thermo Fisher Scientific) in a basic culture medium (BM) consisting of 5% human serum (HS; BioWest, Nuaillé, France) and 1% antibiotics (100 U/mL penicillin; 100 µg/mL streptomycin; Lonza, Basel, Switzerland), in Minimum Essential Medium α, with no nucleosides (MEM α; Thermo Fisher Scientific). The cells were cultured at 37 °C in 5% CO2, detached using TrypLE Select (Thermo Fisher Scientific), and passaged after reaching 70–80% confluency.

hBMSCs were isolated from donated bone marrow aspirates of three female donors aged 81, 87, and 91 obtained during a surgical procedure at the Tampere University Hospital Department of Orthopedics and Traumatology. hBMSC isolation from the aspirates was carried out by filtrating the samples through a cell strainer and centrifugation through a Ficoll gradient (Histopaque^®^-1077; Sigma-Aldrich, Saint Louis, MO, USA), as described by Wang and co-workers [10]. The isolated hBMSCs were cultured like the hASCs, with the exception that BM was supplemented with 5 ng/mL human fibroblast growth factor (hFGF-2; Miltenyi Biotec, Bergisch Gladbach, Germany). 

### 2.2. Characterization of Immunophenotype of the Cells

The immunophenotype of hASC and hBMSC was analyzed by flow cytometry (FACSAria; BD Biosciences, Erembodegem, Belgium) at passage one. Cell samples (10,000 cells/sample) were single stained with the following monoclonal antibodies: CD14-PE-Cy7, CD19-PE-Cy7, CD45RO-APC, CD73-PE, CD90-APC (antibodies from BD Biosciences, Franklin Lakes, NJ, USA), CD11a-APC, CD105-PE (R&D Systems Inc., Minneapolis, MN, USA), CD34-APC, and HLA-DR-PE (Immunotools GmbH, Friesoythe, Germany). A fluorescence level greater than 99% was considered positive.

### 2.3. 3-06 Bioactive Glass Manufacturing and Extract Preparation

The bioactive glass used in this study, an experimental silica-based glass 3-06 of composition [wt%]: Na_2_O 24.6; CaO 21.6; P_2_O_5_ 2.5; B_2_O_3_ 1.3; SiO_2_ 50.0, was manufactured using the melt-quenching method, as described previously [14]. BaG extract medium (BaG BM) was prepared according to the protocol, demonstrated in an earlier study [14]. In brief, 87.5 mg/mL 3-06 granules (500–1000 µm) were disinfected twice using 70% ethanol (Altia, Helsinki, Finland), dried, and then soaked in MEM α supplemented with 1% antibiotics (Lonza) for 24 h at +37 °C to dissolve ions from the glass. After incubation, the extract was sterile filtered, and HS (BioWest) was added to the final concentration of 5%. The BaG medium was stored at +4 °C and used within two weeks.

### 2.4. Cell Seeding and Differentiation Culture

The hASCs and hBMSCs were cultured separately, and the experiments were carried out at passages three to five. The plating density was 1000 cells/cm^2^, except in Western Blot studies 5260 cells/cm^2^. The culture platform for Western Blot samples was Corning Costar^®^ TC-Treated 6 well plate (Corning; Corning, NY, USA), for CyQUANT/ALP, Alizarin Red analyses Nunc^TM^ Cell-Culture Treated polystyrene 24 well plate (Thermo Fisher Scientific), and for immunocytohemical (ICC) and Phalloidin-staining, a chambered polymer coverslip, Ibidi 8 well µ-slide was used (Ibidi GmbH, Gräfelfing, Germany). The cells were seeded into BM. The medium was supplemented with 5 ng/mL hFGF-2 (Miltenyi Biotec) when seeding hBMSCs, but hFGF-2 was excluded from the subsequent media changes.

The osteogenic differentiation was induced 24 h after cell seeding with BaG BM or osteogenic medium (OM) consisting of 5% HS (BioWest), 1% antibiotics (Lonza), 250 µM L-ascorbic acid 2-phosphate, 10 mM β-glyserophosphate, and 5 nM dexamethasone (Sigma-Aldrich) in MEM α (Thermo Fisher Scientific), or with a combination of BaG extract and OM (BaG OM), prepared by supplementing BaG BM with 250 µM L-ascorbic acid 2-phosphate, 10 mM β-glyserophosphate, and 5 nM dexamethasone (the supplements from Sigma-Aldrich). Control cell cultures were maintained in the BM. A total of 3 µM p38 MAPK inhibitor SB202190 (Calbiochem, Merck Millipore, Burlington, MA, USA) was used to inhibit the HSP27 phosphorylation. Fresh media with the inhibitor were changed in the cultures twice a week, during the experiments. 

### 2.5. Western Blot and Immunodetection

hASCs and hBMSCs seeded on 6 well plates (2 donor cell lines of each cell type) were cultured in BM, BaG BM, OM, and BaG OM for 1, 3, 5, 7, 9, and 11 days to create a timeline of the differentiation process. Additionally, hASCs and hBMSCs in BM, BaG BM, OM, and BaG OM supplemented with 3 µM p38 MAPK inhibitor (SB202190; Calbiochem, Merck Millipore, Burlington, MA, USA) were cultured for 7 days.

Cell lysis and Western Blotting was performed as described previously [27]. Briefly, the samples were lysed into Laemmli Sample Buffer (2· concentrate; Bio-Rad, Hercules, CA, USA) supplemented with 5% β-mercaptoethanol (Sigma-Aldrich). The samples were separated on sodium dodecyl sulfate-polyacrylamide gel electrophoresis (SDS-PAGE) gels using electrophoresis and transferred to a polyvinylidene fluoride (PVDF) membrane (Mini Format 0.2 µm; Bio-Rad). The membranes were blocked with a 5% milk blocking solution of 5% skimmed milk powder (Valio, Lapinlahti, Finland) in 0.05% tris-buffered saline (TBS)-Tween (Tween 20; Sigma-Aldrich). Immunodetection was performed using primary antibodies and horseradish peroxidase (HRP) -conjugated secondary antibodies diluted in the blocking solution (listed in Table 1). The antibody incubations were followed by washes with 1· TBS, 0.5% TBS Tween, 0.1% TBS Tween, and 0.05% TBS Tween. The bands were detected with ECL™ Prime Western Blotting Detection Reagent (GE Healthcare, Little Chalfont, UK), and Chemi Doc MP System (Bio-Rad). Semi-quantitative analysis of immunoblotted pHSP27(S78) and β-actin protein amounts was performed with Image J [39] to show the normalized pHSP27(S78) protein level. 

### 2.6. Cell Proliferation and Quantitative Analysis of Alkaline Phosphatase Activity

Cell proliferation, i.e., total DNA amount of hASCs and hBMSCs, was determined quantitatively by CyQUANT^®^ cell proliferation assay (Thermo Fisher Scientific) after 7, 9, and 11 days of culture. Before cell lysis, time point images were taken with a Nikon eclipse TS100 inverted phase contrast microscope (Nikon, Tokyo, Japan), with 4 X light objective. The analysis was performed on samples collected in 0.1% Triton buffer (Sigma-Aldrich) according to the manufacturer’s protocol, as described previously [38,40]. The fluorescence signal representing nucleic acid bound CyQUANT^®^ GR dye (Thermo Fisher Scientific) was measured at 480/520 nm with a microplate reader (Victor 1420 Multilabel Counter, Wallac, Turku, Finland). 

Alkaline phosphatase activity denoting the early osteogenic differentiation of hMSCs was analyzed from the same cell lysates as cell proliferation, as previously described [38,40]. Briefly, the samples were incubated 15 min at 37 °C in working solution consisting of 1:1 10.8 µM phosphatase substrate (Sigma-Aldrich), and 1.5 M alkaline buffer solution (Sigma-Aldrich), after which the reaction was halted with 1.0 M sodium hydroxide (Sigma-Aldrich). The chromogenic reactions of the samples were detected by measuring the absorbance at 405 nm with a Victor 1420 microplate reader (Wallac). 

### 2.7. Alizarin Red Staining of Matrix Mineralization

Extracellular mineral formation was studied with Alizarin Red (AR) analysis after 9 and 11 days of differentiation culture, as reported before [27]. In brief, the wells were fixed with 70% ethanol (Altia), stained with Alizarin Red S solution, (pH 4.1–4.3; Sigma-Aldrich) for 10–15 min, washed with water and then with 70% ethanol (Altia). Dry wells were photographed for macroscopic images, and AR staining was quantified by extracting the dye for 3 h into 100 mM cetylpyridinium chloride (Sigma-Aldrich). The absorbances of the extracted dye were measured at 544 nm with a Victor 1420 microplate reader (Wallac). 

### 2.8. Immunocytochemical Staining

ICC staining of hASCs and hBMSCs (2 donor cell lines of each cell type) was conducted to analyze the protein expression of HSP27 (time point 9 d) and pHSP27(S78) (time points 3 d and 9 d), and osteogenic marker protein collagen type I, at 11 days of culture. ICC staining was performed as described previously [27]. The cell cultures were fixed for 15 min with 0.2% Triton X-100 in paraformaldehyde (PFA; Sigma-Aldrich) and blocked with 1% bovine serum albumin (BSA; Sigma-Aldrich) in phosphate buffered saline (PBS; Lonza). The samples were incubated with primary antibodies, washed, and incubated with secondary antibodies (detailed antibody and reagent information is given in Table 2). Phalloidin-Tetramethylrhodamine B isothiocyanate (TRITC), for actin cytoskeleton staining, was added simultaneously with the secondary antibodies. The samples were counterstained with 4’,6-diamidino-2-phenylindole (DAPI) for visualization of nuclei. Secondary antibody control staining was conducted as described, with the exception that the primary antibody incubation was left out.

### 2.9. Fluorescence Imaging and Image-Based Analysis of Stain Intensity

Fluorescence images of hASC and hBMSC samples were taken with an Olympus IX51 inverted microscope (Olympus, Tokyo, Japan) equipped with a fluorescence unit and camera (DP30BW). Alexa 488 filter was used for detection of HSP27, pHSP27(S78), and collagen type I, an Alexa 546 filter for β-actin, and a DAPI filter for nuclei. The samples were imaged with 40 X objective. Exposure time was kept constant within experiments. Image panels were processed, and adjustments of brightness and contrast were made with Adobe Photoshop CC (Adobe, San Jose, CA, USA). The 4 X digital zoom images were processed with Fiji [41]. The mean grey values representing the intensity of HSP27, pHSP27, and Phalloidin-TRITC signals in the fluorescence images, and the nuclei count of corresponding images for normalization of the intensities, were analyzed with Fiji [41]. 

### 2.10. Statistical Analysis

Statistical analyses were performed to evaluate the differences between samples in the ALP, CyQUANT, and Alizarin Red analyses, and the image-based quantitation data of HSP27, pHSP27, and F-actin. Results are expressed as mean and standard deviation (SD). Statistical analyses were conducted using the non-parametric Mann–Whitney test, followed by the Bonferroni post-hoc test with GraphPad Prism 5 (La Jolla, CA, USA). Statistical differences with *p* ˂ 0.05 were considered significant. BM BaG, OM, and OM BaG conditions were compared to BM control. Comparisons of BM BaG vs OM BaG, OM vs OM BaG, and the comparisons between BM, BaG BM, OM, and BaG OM, and the corresponding SB202190-inhibited conditions were also made due to their relevance.

## 3. Results

### 3.1. Characterization of the Mesenchymal Origin of the CELLS

The cells were identified as mesenchymal based on the criteria given by the International Society for Cellular Therapy [42]. The cells were adherent to plastic culture surfaces and the surface marker expression pattern studied by flow cytometry conveyed the criteria. All hASC and hBMSC donor cell lines in this study had positive expression of CD73, CD90, and CD105, and lacked the expression of CD11, CD14, CD19, and CD45 (Table 3). The expression of CD34 was negative in hBMSCs, but moderate in hASCs. The elevated expression of CD34 in freshly isolated hASCs declines after passaging [43]. HLA-DR expression of hASCs was negative and in accordance with the criteria, but hBMSC donor cell lines were HLA-DR positive. This feature of BMSCs has been reported under normal culture conditions and does not compromise the MSC identity [44]. 

### 3.2. Proliferation and Osteogenic Differentiation

The CyQUANT assay results of cell proliferation (Figure 1A,B) show, that BaG extract enhanced the total DNA amount of both hASCs and hBMSCs, when cultured in BM. In hASCs, the cell number was increased with OM, at all time points studied. Compared to the OM condition, BaG OM significantly enhanced cell proliferation of hASCs on day 7, but diminished the cell number on day 11. Proliferation was significantly induced with OM in hBMSCs, but BaG OM reduced the proliferation compared with the OM condition. Micrographs taken before the sample lysis show that the cells cultured under BaG OM for 9 and 11 days had produced a mineral layer on top of the cell layer (Appendix A). 

We evaluated the osteogenic potential of hASCs and hBMSCs by analyzing the activity of an early osteogenic marker, alkaline phosphatase (ALP), normalized with the corresponding cell amount (Figure 1C,D). The ALP activity of hASCs was time-dependently increased with OM and BaG OM conditions. OM induced a significant increase in ALP activity of hASCs on day 11, which was further enhanced by BaG OM. OM significantly increased the ALP activity of hBMSCs at all time points studied. Furthermore, BaG OM stimulated a significant increase in the ALP activity of hBMSC already on day 7, compared with OM alone. ALP activity of both cell types was low under BM and BaG BM conditions.

Mineral accumulation in the extracellular matrix (ECM), characteristic of late osteogenic differentiation, was analyzed by Alizarin Red (AR) staining of hASCs and hBMSCs, after 9 and 11 days of osteogenic differentiation (Figure 1E,F). Based on the quantitative AR staining and corresponding qualitative images shown below the columns, BaG OM significantly enhanced matrix mineralization of both hASCs and hBMSCs. OM alone induced mineralization compared with the BM control, but the level of mineral accumulation was markedly lower without BaG. Only hBMSCs demonstrated strong red AR staining in OM, on day 11. BaG without osteogenic supplements could not stimulate mineralization of the ECM during the culture period. 

To further analyze the osteogenic potential of the hASCs and hBMSCs, the production of ECM protein collagen type I was analyzed with ICC staining on day 11 (Figure 1 G,H, secondary antibody controls in Appendix A). In hASCs, OM enhanced the collagen type I production, which appeared to localize intracellularly in the perinuclear area. BaG OM markedly enhanced the amount of collagen type I in hASCs, which was found secreted into the extracellular space. In hBMSCs, extracellular fibrils of collagen type I were formed under OM condition, but there was more collagen type I present in BaG OM. The production of collagen type I was negligible under the BM and BaG BM conditions. 

### 3.3. Activation of the p38/MK2/HSP27 Pathway 

The activation of p38/MK2/HSP27 pathway during hASC and hBMSC osteogenesis was analyzed with Western Blotting. The timeline figures (Figure 2A,B and Appendix A presenting additional donor cell lines) show the sequential activation of p38 MAPK, MK2, and HSP27 through phosphorylation, and the corresponding unphosphorylated protein levels. The pathway constituents were detected in hASCs and hBMSCs cultured under all culture conditions, but the protein expressions were upregulated more in OM conditions. The protein levels of p-p38, p38, pMK2, MK2, and pHSP27(S78) increased with time during culture in OM, whereas in BaG OM, the expression levels peaked and declined during the culture. Unphosphorylated HSP27 and β-actin were present throughout the culture period, under all culture conditions. Semi-quantification of the pHSP27(S78) normalized with the β-actin, representing the cell amount, was performed to highlight its activation (Figure 2C,D). The normalized pHSP27(S28) peaked later in hASCs compared to hBMSCs, under BaG OM condition. In hBMSCs the normalized pHSP27(S28) increased time-dependently in OM without BaG, while a similar trend was absent in the hASCs. 

### 3.4. Inhibition of HSP27 Phosphorylation

The relevance of pHSP27(S78) activation in BaG induced osteogenesis of hASCs and hBMSCs was further studied with p38 MAPK inhibition, under OM and BaG OM conditions. SB202190 inhibitor with a concentration of 3 µM was chosen based on the literature and prior optimization [45,46]. SB202190 was shown by Western Blotting to specifically reduce the phosphorylation, and thus activation of pHSP27(S78), without affecting the basal level of HSP27 (Appendix A). CyQUANT analysis was performed to confirm that the inhibitor did not compromise cell proliferation (Appendix A). 

The p38 MAPK inhibition had a reducing trend on BaG OM -induced ALP activity (Figure 3A,B). SB202190 significantly decreased the BaG OM induced mineralization of the ECM in both cell types on day 11 (Figure 3C,D), although mineralization was not fully inhibited. Similarly, ICC staining showed that inhibition with SB202190 diminished collagen type I production of hASCs, under OM and BaG OM conditions (Figure 3E,F). In hBMSCs, the level of collagen type I staining was diminished by p38 MAPK inhibition in OM, but there was no clear difference in BaG OM. 

### 3.5. HSP27 and pHSP27 Localization

Cellular localization of HSP27 and its phosphorylated form, cultured under BaG and SB202190 treatment, were visualized with ICC staining, and the stain intensities were quantified and normalized with the cell number. After 9 days of culture, ICC stained basal HSP27 was scarce under BM and BaG BM conditions (Appendix A), but found abundantly in the cytosol of hASCs and hBMSCs cultured, under OM and BaG OM conditions (Figure 4A). The quantitative data showed that p38 MAPK inhibition had a reducing effect on HSP27 in hASCs, when cultured under OM and BaG OM, whereas the inhibition had no effect in hBMSCs (Figure 4B,C).

ICC staining of pHSP27(S78) after 3 days of differentiation showed that when phosphorylated at Serin 78, HSP27 co-localized with the rigid and aligned Phalloidin-stained F-actin fibers (Figure 5A and Appendix A). Similarly on day 9, pHSP27(S78) associated with the actin filaments in hASCs and hBMSCs, cultured under BM, BaG BM, and OM conditions (Figure 5D and Appendix A). Interestingly, pHSP27(S78) was found diffusely located throughout the cytosol after 9 days of differentiation in BaG OM, in both cell types. The diffuse localization of pHSP27(S78) was also partly present in hBMSCs cultured in OM. The p38 MAPK inhibition on day 9 reversed the BaG OM related diffuse localization of pHSP27(S78) to more F-actin aligned organization (Figure 5D). The image-based quantitation of pHSP27(78) stain intensity showed that p38 MAPK inhibition reduced the normalized amount of the phosphoprotein, especially in the later time point (Figure 5E,F).

### 3.6. The Cell Morphology and F-Actin Intensity 

The role of HSP27 phosphorylation on the cytoskeletal organization was further analyzed with Phalloidin-TRITC staining and image-based quantitation of F-actin intensity. After 9 days of differentiation, the cell morphology of hASCs and hBMSCs (Figure 6A), was fibroblastic with thick stress fibers traversing the cells, in the BM and OM conditions. BaG extract or p38 MAPK inhibitor treatment altered the morphology leading to thinner and less visible F-actin fibers. The osteogenically committed hMSCs in BaG OM condition had markedly different appearance with disrupted cytoskeleton and some disintegrated nuclei. Quantitation of the imaging data revealed that the mean F-actin intensity, normalized with cell amount (fragmented nuclei were excluded), was significantly diminished by BaG extract addition in both cell types after 9 days of differentiation (Figure 6B,C). Similarly, p38 MAPK inhibition significantly reduced the mean F-actin intensity in all other conditions except BaG OM in hBMSCs, where the amount of F-actin was already low without the inhibition.

## 4. Discussion

Bioactive glasses and their combination with stem cells are considered promising remedies for bone defects, although the molecular details on how BaGs induce osteogenesis remain understudied. The importance of the regulation of the actin cytoskeleton in the osteogenic commitment of MSCs has been demonstrated in several studies [19,24,26,27,28,47]. MAPK p38 downstream substrate, a chaperone HSP27, has been proposed as a regulator of cytoskeletal dynamics, but its role in osteogenesis is not established. We hypothesized that HSP27 driven cytoskeletal modulation would be linked to the osteogenic course of hMSCs. The efforts made in unraveling the signaling pathways involved in BaG induced osteogenesis are important for creating knowledge-based biomaterials for bone regeneration.

We differentiated hMSCs derived from bone marrow and adipose tissue to make our data representative of hMSCs in BaG extract medium (BaG BM), osteogenic culture medium (OM), or with a combination of BaG extract and osteogenic supplements (BaG OM). BaGs have been found to be strong inducers of osteogenic differentiation of hMSCs [6,8], and our research group has confirmed that solely the ions extracted from bioactive glasses can enhance osteogenesis of hASCs [13,14]. The experimental BaG used in this study was a S53P4-based BaG with the substitution of silica with boron, shown to have a fast glass dissolution rate and osteogenic response in hASCs [14], and a good bone attachment in a rat model [48]. Indeed, boron has been demonstrated as an important trace element in bone formation [9,49,50,51,52]. 

In accordance with our previous publications [13,14], we demonstrated here that ionic dissolution of the 3-06 BaG supplemented with osteogenic agents induced significant and rapid osteogenic responses in hMSCs harvested from adipose tissue and bone marrow, as demonstrated with enhanced ALP activity and mineralization of the ECM, already on day 9 of differentiation. The osteogenic response of the cells was accompanied with a decrease in cell proliferation, as shown previously [53,54], but no toxic effects from the ionic extract were observed. The strong mineralization of hMSCs as a response to the 3-06 BaG was in coherence with the earlier borosilicate glass studies [51,55]. The ALP activity and mineralization also elevated with time in OM without BaG, suggesting to delayed osteogenic response without the ionic stimulus.

Collagen type I is the major component of the bone ECM, accounting for 90% of the organic matrix [56], and it is linked to osteogenic maturation [57]. Here we showed that collagen type I production was enhanced by osteogenic condition, and further increased by the ionic extract of 3-06 BaG. The collagen network in the ECM provides a platform for calcium phosphate -based crystal formation, and thus mineralization of the bone matrix [58,59,60]. The strong collagen type I staining induced by BaG OM was in coherence with the mineralization assay, indicating late-stage osteogenesis of the hMSCs. 

BaGs have been reported to activate osteogenesis even without osteogenic culture supplements [15,61]. In this study, the ionic dissolution of 3-06 bioactive glass in basic medium did not elicit osteogenesis. Supporting these findings, in our previous study, the ionic dissolution products of BaGs alone could not induce the osteogenic response of hASCs [14]. However, the culture period in this study could have been simply too short, or the ionic cocktail insufficient, for BaG induced osteogenesis without other culture supplements. 

The expression of HSP27 is upregulated as a response to conditions that alter protein folding, such as heat shock [31], but also mechanical stimuli [17], and differentiation [32]. In an earlier study with hBMSC, HSP27 gene expression was upregulated with electrical stimulation starting on day 10, and the osteogenic differentiation markers ALP and collagen type I were upregulated shortly after [33]. Additionally, in hASCs, HSP27 protein activity was upregulated with osteogenic differentiation medium [62]. We analyzed the protein expression of the p38/MK2/HSP27 pathway constituents p38, MK2, and HSP27, and their phosphorylated forms, and found coherently that the signaling axis was activated with osteogenic induction. A 3-06 BaG extract induced strong, transient HSP27 phosphorylation at Ser-78, which was temporarily coupled to the activation of upstream kinases p38 MAPK and MK2, by phosphorylation. The early activation of signaling seemed to be linked to the fast osteogenic response of hASCs and hBMSCs, under BaG OM. Supporting our findings, the ionic products of Bioglass 45S5 have been reported to enhance p38 MAPK and MK2 gene expression in osteoblasts [16]. The cells under OM showed early markers of osteogenesis and enhanced activation of p38/HSP27 signaling throughout the culture period, suggesting its role in the osteogenic commitment.

The role of pHSP27(S78) in directing the osteogenic course of hMSCs was further assessed by an upstream kinase inhibition. The pharmacological inhibition of HSP27 has been previously conducted with p38 MAPK inhibitors SB202190, and a structurally related SB203580 (Adezmapimod) [29,46,63]. Here, we confirmed that SB202190 specifically inhibited phosphorylation of HSP27 at Ser-78 in hBMSCs and hASCs. Under inhibitor treatment, the BaG induced ALP activity was slightly reduced, and the matrix mineralization was significantly reduced, although not fully inhibited. p38 MAPK inhibition also hampered the production of collagen type I. Our results suggest that HSP27 phosphorylation is involved in the osteogenic commitment of hMSCs, but presumably is not a determining factor of differentiation fate. 

The cytoskeletal association of HSP27 has been demonstrated in earlier studies, but the role of the HSP27 phosphorylation status in the cytoskeletal localization remains contradictory [17,29,36,64]. In this study, the localization of unphosphorylated HSP27 was cytoplasmic, similar to previously described results [29,65]. HSP27 co-localized with actin only when phosphorylated, forming linear structures. Although HSP27 has been more commonly described as an actin capping protein [30], it may also bind to the sides of actin filaments as monomers [66,67], as in our observations. Similar results were reported in mouse fibroblasts, where uniaxial cyclic stretch upregulated HSP27 phosphorylation and subsequent co-localization of pHSP27 with actin stress fibers [17]. This cytoskeletal organization of pHSP27 was shown to reinforce the actin filaments [17]. In this study, the inhibition of p38/HSP27 axis reduced the pHSP27(S78) protein level and had a parallel effect on the F-actin intensity, suggesting a stabilizing function of pHSP27 on the actin filaments. 

Studies have demonstrated that actin polymerization drives the MSC differentiation towards an osteogenic fate [19,68]. We discovered that the cytoskeletal association of pHSP27(S78) was linked to the undifferentiated or to the early osteocommitted hMSCs state with prevalent actin cytoskeleton, and was not present in the cells, strongly demonstrating mineralization of the collagenous matrix. Indeed, the cytoskeletal rigidity and tension are cues of the osteogenic commitment, but the differentiation process involves dynamic remodeling of the actin cytoskeleton [19,24,28]. Our results suggest that phosphorylation of HSP27 at Ser-78 could perform as an early regulatory step in promoting the hMSCs to commit to the osteogenic course through its interaction with the actin filament cytoskeleton. However, it is possible that this regulation involves a combination of multiple signaling molecules downstream of p38 MAPK.

## 5. Conclusions

In this study, we investigated the currently unstudied role of the p38 MAPK substrate HSP27 in hMSC BaG induced osteogenesis. We discovered that the ionic extract of an experimental silica-based 3-06 BaG, together with osteogenic culture supplements, induced rapid osteogenesis in hMSCs of two different origins, demonstrated by enhanced ALP activity, matrix mineralization, and secretion of collagen type I. The protein expression and activation by phosphorylation of p38 MAPK, MK2 and HSP27(S78) was linked to the osteogenic outcome, and the inhibition of HSP27 phosphorylation reduced the osteogenic markers, indicating the importance of this pathway to hMSC osteogenic course. Co-localization of pHSP27(S78) with actin stress fibers was linked to the early osteogenesis of hMSCs, whereas its localization was diffuse in the differentiated hMSCs. Due to the interplay of p38 MAPK signaling with several other targets, further studies on the topic are required.

## Figures and Tables

**Figure 1 cells-12-00224-f001:**
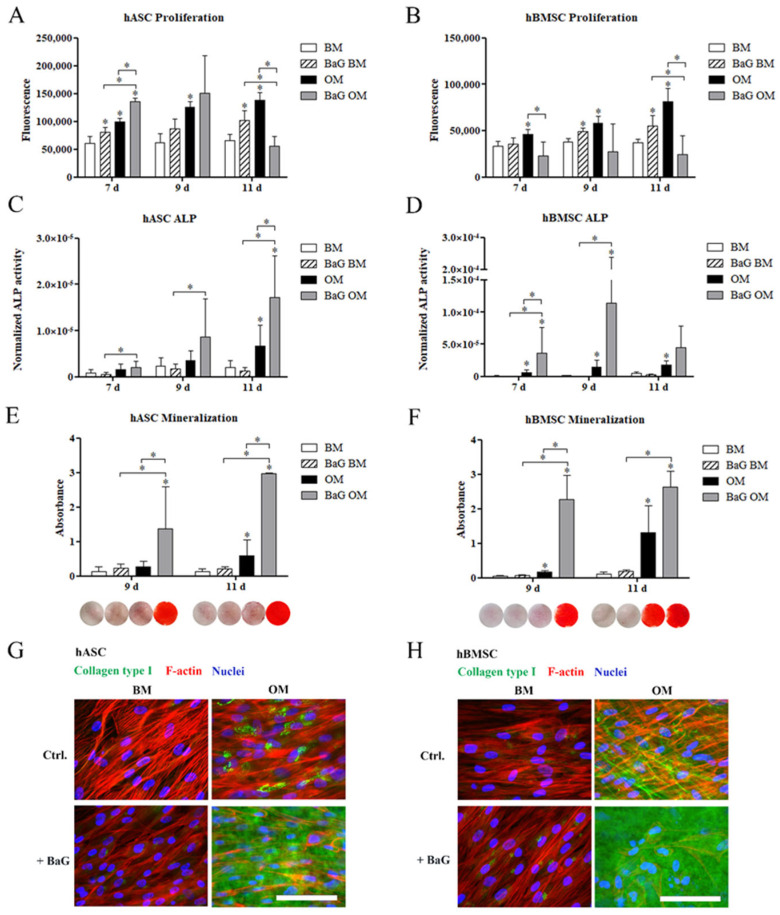
Cell proliferation and osteogenic differentiation of hASCs and hBMSCs. hASCs and hBMSCs were cultured in BM, BaG BM, OM, and BaG OM. (**A**,**B**) Cell proliferation was studied with CyQUANT assay after 7, 9, and 11 days of culture. (**C**,**D**) ALP activity was studied after 7, 9, and 11 days of culture and is presented normalized, with the corresponding cell amounts. (**E**,**F**) Matrix mineralization was analyzed with Alizarin Red staining after 9 and 11 days of culture. Qualitative representative results (stained wells, area 1.9 cm^2^) are presented below the corresponding graphs. Bright red staining represents the mineral. (**G**,**H**) Representative images of collagen type I stained hASCs and hBMSCs at 11 d. Scale bars 100 µm. Statistical analysis: cell proliferation and ALP analyses; N = 9, independent biological replicates of three donors of each cell type. Mineralization; hASCs: N = 9 independent biological replicates of three donors; and hBMSCs: N = 6 independent biological replicates of two donors. Statistical analysis was conducted within time point, * *p* ˂ 0.05. The asterisk above the error bar indicates the statistical difference compared to the BM control, other comparisons are indicated with lines. Abbreviations: alkaline phosphatase, ALP; basic medium, BM; osteogenic medium, OM; and bioactive glass, BaG.

**Figure 2 cells-12-00224-f002:**
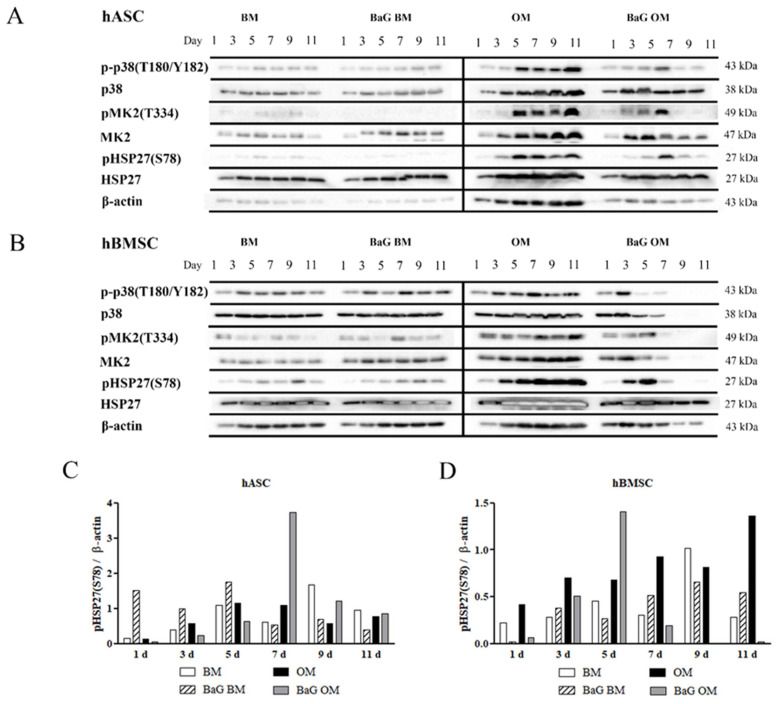
Timeline of p38/MK2/HSP27 activation in hASCs and hBMSCs. (**A**,**B**) hASCs and hBMSCs were cultured in BM, BaG BM, OM, and BaG OM for 1, 3, 5, 7, 9, and 11 days, and analyzed with Western Blotting and immunodetection. The figure presents cropped blots of p-p38, p38, pMK2, MK2, pHSP27(S78), HSP27, and β-actin of the representative donor cell lines. Full-length blots are presented in Appendix A. (**C**,**D**) Semi-quantification of the pHSP27(S28) bands presented normalized to β-actin in hASCs and hBMSCs. Abbreviations basic medium, BM; osteogenic medium, OM; and bioactive glass, BaG.

**Figure 3 cells-12-00224-f003:**
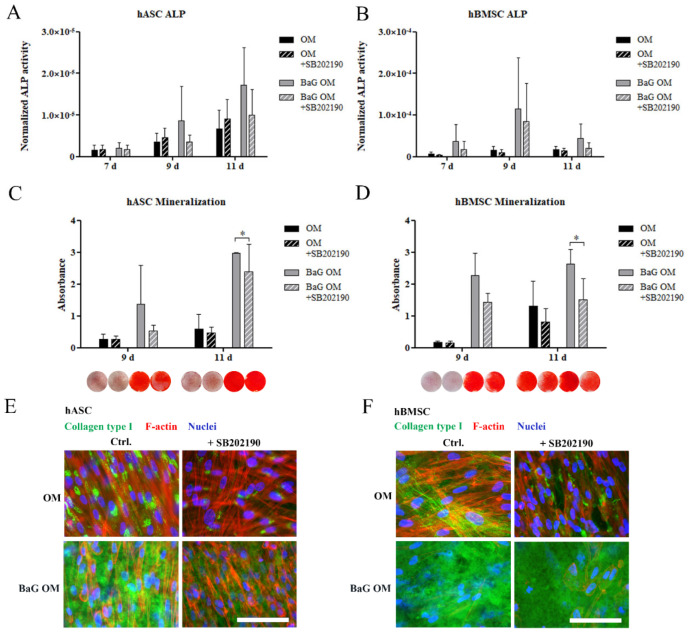
The effect of p38 MAPK inhibition with SB202190 on hASC and hBMSC osteogenesis. hASCs and hBMSCs were cultured in BM, BaG BM, OM, and BaG OM supplemented with 3 µM SB202190 inhibitor. (**A**,**B**) Early osteogenic differentiation was assessed with ALP activity. (**C**,**D**) Matrix mineralization was analyzed with Alizarin Red staining (stained wells, area 1.9 cm^2^). (**E**,**F**) ICC staining of collagen type I. Scale bars are 100 µm. Values for OM and BaG OM conditions without the inhibitor, and qualitative representative results of Alizarin Red staining are the same as presented in Figure 1. Statistical analysis: ALP analysis (and cell proliferation used to normalize the ALP activity data); N = 9, independent biological replicates of three donors of each cell type. Mineralization; hASCs: N = 9 independent biological replicates of three donors; hBMSCs: N = 6 independent biological replicates of two donors. Statistical analysis was conducted within the time point, and comparisons were made between OM and OM + SB202190 or BaG OM and BAG OM + SB202190. ** p* ˂ 0.05. Abbreviations: osteogenic medium, OM; and bioactive glass, BaG.

**Figure 4 cells-12-00224-f004:**
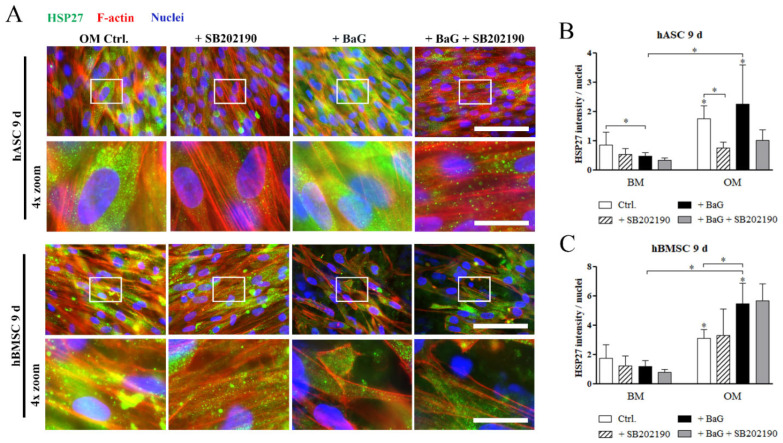
Cellular localization of basal HSP27 in hASCs and hBMSCs. (**A**) Representative images of HSP27-stained hASCs and hBMSCs after 9 days of culture in OM, and BaG OM supplemented with 3 µM SB202190 inhibitor. Fluorescence images were taken with a 40· magnification and with filters for HSP27 (Alexa 488, green), actin (Alexa 546, red), and nuclei (DAPI, blue). The 4 X digital zoom images are presented below the corresponding original images, indicated with white rectangles. Secondary antibody controls are presented in Appendix A. Scale bars are 100 µm in the original images and 25 µm in the zoom images. (**B**,**C**) Image-based quantification of mean grey values of HSP27 stained hASCs and hBMSCs normalized with nuclei count. Statistical analysis: hASCs: N = 8–9 images, 2 independent biological replicates of 2 donors; hBMSCs: N = 7–8 images, 2 independent biological replicates of 2 donors. Representative images of hASCs and hBMSCs cultured in BM and BaG BM conditions are in Appendix A. ** p* ˂ 0.05. The asterisk above the error bar indicates the statistical difference compared to the BM control, and other comparisons are indicated with lines. Abbreviations: basic medium, BM; osteogenic medium, OM; and bioactive glass, BaG.

**Figure 5 cells-12-00224-f005:**
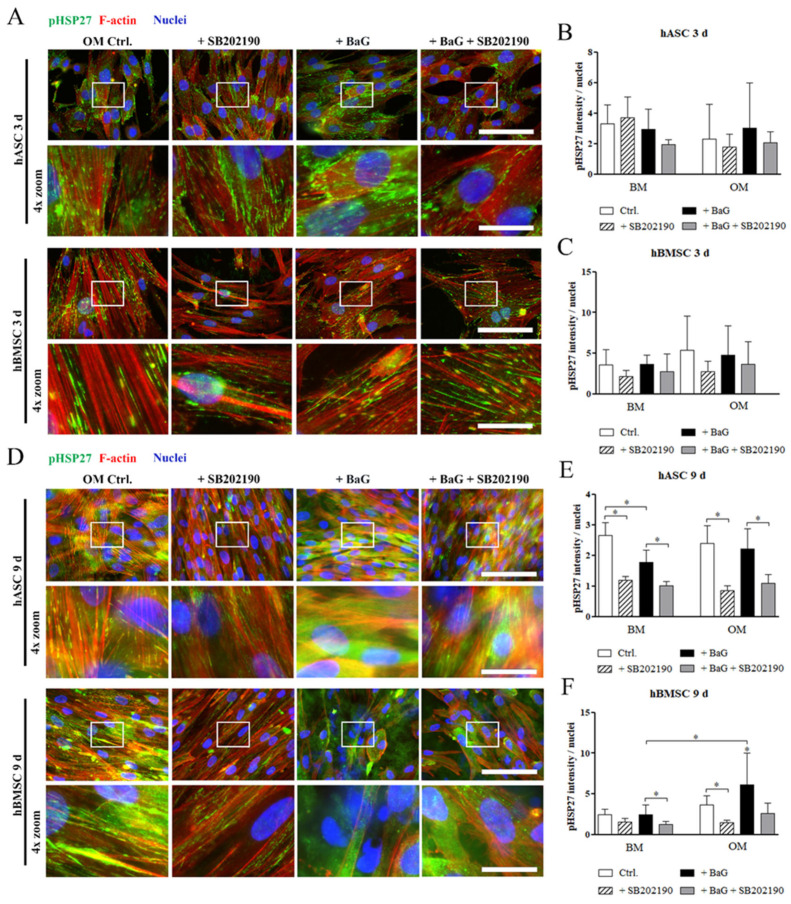
Cellular localization of phosphorylated HSP27 in hASCs and hBMSCs. (**A**,**D**) Representative images of pHSP28(S78) stained hASCs and hBMSCs, after 3 and 9 days of culture in OM and BaG OM. Fluorescence images were taken with a 40 X magnification and with filters for pHSP27(S78) (Alexa 488, green), actin (Alexa 546, red), and nuclei (DAPI, blue). The 4 X digital zoom images are presented below the corresponding original images, indicated with white rectangles. Secondary antibody controls are presented in Appendix A. Scale bars are 100 µm in the original images and 25 µm in the zoom images. (**B**,**C**,**E**,**F**) Image-based quantification of mean grey values of pHSP27 stained hASCs and hBMSCs normalized with nuclei count. Statistical analysis: hASCs: N = 6–9 (3 d)/8–10 (9 d) images, 2 independent biological replicates of 2 donors; hBMSCs: N = 7–9 (3 d)/7–11 (9 d) images, 2 independent biological replicates of 2 donors. Representative images of hASCs and hBMSCs cultured in BM and BaG BM conditions are in Appendix A. ** p* ˂ 0.05. The asterisk above the error bar indicates the statistical difference compared to the BM control, and other comparisons are indicated with lines. Abbreviations: basic medium, BM; osteogenic medium, OM; and bioactive glass, BaG.

**Figure 6 cells-12-00224-f006:**
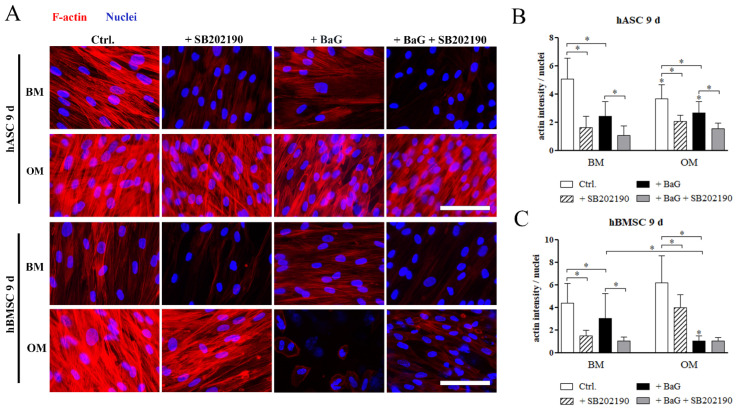
Actin cytoskeleton and F-actin intensity of hASCs and hBMSCs. (**A**) Representative Phalloidin-TRITC stained hASCs and hBMSCs cultured 9 days in BM, BaG BM, OM, and BaG OM supplemented with 3 µM SB202190 inhibitor. The cells were imaged with a fluorescence microscope using constant exposure times with Alexa 546 for actin (red) and DAPI (blue) filters, and a 40× magnification. Scale bar is 100 µm, and is used in every image. (**B**,**C**) Image-based quantification of mean grey values of Phalloidin-TRITC stained hASCs and hBMSCs normalized with nuclei count. Statistical analysis: hASCs: N = 16–18 images, 4 independent biological replicates of 2 donors; hBMSCs: N = 15–18 images, 4 independent biological replicates of 2 donors. ** p* ˂ 0.05. The asterisk above the error bar indicates the statistical difference compared to the BM control, and other comparisons are indicated with lines. Abbreviations: basic medium, BM; osteogenic medium, OM; and bioactive glass, BaG.

**Table 1 cells-12-00224-t001:** Primary and secondary antibodies used in immunodetection.

Antibody Type	Antibody	Host Species	Dilution	Incubation
Primary	anti-β-actin sc-47778 _†_	mouse	1:2000	RT, 2 h
Primary	anti-HSP27 (D6W5V) #95357S _‡_	rabbit	1:1000	+4 °C, overnight
Primary	anti-p-HSP27(S78) #2405S _‡_	rabbit	1:1000	+4 °C, overnight
Primary	anti-MAPKAPK2 #12155T _‡_	rabbit	1:1000	+4 °C, overnight
Primary	anti-p-MAPKAPK2 (T334) #3007T _‡_	rabbit	1:1000	+4 °C, overnight
Primary	anti-p38α sc-728 _†_	rabbit	1:100	+4 °C, overnight
Primary	anti-p-p38 MAPK(T180/Y182) #4511S _‡_	rabbit	1:1000	+4 °C, overnight
Secondary	anti-mouse IgG-HRP (sc-2005) _†_	goat	1:2000	RT, 1h
Secondary	anti-rabbit IgG-HRP #7074S _‡_	goat	1:2000	RT, 1h

_†_ Santa Cruz Biotechnology, Dallas, TX, USA. _‡_ Cell Signaling Technology, Danvers, MA, USA.

**Table 2 cells-12-00224-t002:** Reagents used in ICC staining.

Antibody Type	Antibody	Host Species	Dilution	Incubation
Primary	anti- collagen type I (ab90395) _†_	mouse	1:2000	+4 °C, overnight
Primary	anti-pHSP27 (S78) (ab32501) _†_	rabbit	1:500	+4 °C, overnight
Primary	anti-HSP27 (D6W5V) #95357S _‡_	rabbit	1:500	+4 °C, overnight
Secondary	anti-mouse IgG Alexa fluor 488 (A11029) _§_	goat	1:500	+4 °C, 45 min
Secondary	anti-rabbit IgG Alexa fluor 488 (A21206) _§_	donkey	1:500	+4 °C, 45 min
-	Phalloidin-TRITC _§_	-	1:500	+4 °C, 45 min
-	DAPI _§_	-	1:2000	RT, 5 min

_†_ Abcam, Cambridge, United Kingdom. _‡_ Cell Signaling Technology, Danvers, MA, USA. _§_ Sigma-Aldrich, Saint Louis, MO, USA.

**Table 3 cells-12-00224-t003:** Surface protein expression of hASCs and hBMSCs.

Antigen	Surface Protein	Surface Marker Expression hASC (%)	Surface Marker Expression hBMSC (%)	Fluorophore	Manufacturer
CD11a	Integrin alpha L (Lymphocyte function-associated antigen 1)	1.5 ± 0.7	0.7 ± 0.4	APC	R&D Systems Inc. Minneapolis. MN. USA
CD14	Lipopolysaccharide receptor	1.2 ± 0.9	5.4 ± 1.8	PECy7	BD Biosciences. Franklin Lakes. NJ. USA
CD19	B lymphocyte-lineage differentiation antigen	1.1 ± 0.9	4.3 ± 2.4	PECy7	BD Biosciences
CD34	Hematopoietic progenitor cell antigen 1	22.2 ± 23.4	2.5 ± 0.5	APC	Immunotools GmbH. Friesoythe. Germany
CD45	RO isoform of leucocyte common antigen	1.8 ± 0.3	7.1 ± 1.4	APC	BD Biosciences
CD73	Ecto-5′-nucleotidase	91.1 ± 3.7	93.4 ± 3.7	PE	BD Biosciences
CD90	Thy-1 (T cell surface glycoproteins)	98.8 ± 1.0	88.6 ± 6.5	APC	BD Biosciences
CD105	SH-2. Endoglin	94.7 ± 6.9	92.9 ± 6.1	PE	R&D Systems Inc.
HLA-DR	Major histocompatibility class II antigen (MHC-II)	1.3 ± 1.1	85.7 ± 8.3	PE	Immunotools GmbH

Abbreviations: human adipose stem cell, hASC; human bone marrow stem cell, hBMSC; cluster of differentiation, CD; allophycocyanin, APC; phycoerythrin-cyanine, PECy7; phycoerythrin, PE. Standard deviation is indicated with ±.

## Data Availability

The datasets used and analysed during the current study are available from the corresponding author on reasonable request.

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
