# Peer review of "Heat Shock Protein 27 Is Involved in the Bioactive Glass Induced Osteogenic Response of Human Mesenchymal Stem Cells"

_cells, 2023, doi:10.3390/cells12020224_

Round 1
Reviewer 1 Report
In the present manuscript, Hyväri et al. studied a potential novel mechanism of p38 downstream signaling in the bioactive glass induced osteogenic response of hMSCs. The introduction describes the scientific background and the aim of the study in an appropriate manner. A detailed description of the methods and results helps the reader to understand the topic. In summary, this is a well written and interesting manuscript. However, from my point of view some additional experiments are missing to underline the authors conclusion, that BaG induces osteogenic differentiation via p38 and HSP27. Please find below my points of concern.
- All figures should be increased in size.
- In ref. 14, the authors measured the ion concentrations of the BaG extracts. This information should be included in the introduction to better guide the reader. Moreover, it should be explained why the authors used BaG extracts and not the BaG itself.
- In Figure 1 G,H and ll.287-291 the authors refer to the amount of extracellular and intracellular collagen type I. Based on the provided immunocytochemical stainings it is difficult to decide which fluorescence signal is extracellular and which is intracellular. Is there a possibility to quantify the amount of extracellular vs. intracellular collagen type I?
- The authors are encouraged to describe the effect of p38 inhibition on the reorganization of actin filaments and the localization of HSP27 an pHSP27 in additional experiments
- A quantification of the western blots in Figure 2 is missing. The signal of the proteins of interest should be normalized to ß-actin levels.
- How do the authors explain that hASCs are more inducible by BaG than hBMSCs (Figure 4)? This could be included in the discussion.
- Figure 4: It would be interesting to discuss how the diffuse localization of pHSP27 in BaG OM affects the osteogenic differentiation and bone repair.
- The authors are encouraged to perform knockdown of HSP27 (potentially with RNAi) and subsequent osteogenic differentiation to validate the hypothesis that HSP27 is essential in the BaG-induced osteogenic differentiation.
Reviewer 2 Report
Please see the attachment.

Round 2
Reviewer 1 Report
The revised manuscript of Hyväri and colleagues provides new data which strengthen the authors conclusions. Moreover, the extension of the introduction and discussion sections help to understand the aim of the study and its conclusions. From my point of view the quality of the manuscript has improved significantly.
Reviewer 2 Report
The authors have sufficiently addressed my specific concerns.